# Targeting the Interplay between Cancer Metabolic Reprogramming and Cell Death Pathways as a Viable Therapeutic Path

**DOI:** 10.3390/biomedicines9121942

**Published:** 2021-12-18

**Authors:** Elisabetta Iessi, Rosa Vona, Camilla Cittadini, Paola Matarrese

**Affiliations:** Center for Gender-Specific Medicine, Istituto Superiore di Sanità, Italian National Institute of Health, 00161 Rome, Italy; elisabetta.iessi@iss.it (E.I.); rosa.vona@iss.it (R.V.); camilla.cittadini@iss.it (C.C.)

**Keywords:** cancer cell metabolism, cell death, anticancer therapy, chemoresistance, glucose, glycolysis, acidity, oxidative metabolism, OXPHOS, tumor microenvironment

## Abstract

In cancer cells, metabolic adaptations are often observed in terms of nutrient absorption, biosynthesis of macromolecules, and production of energy necessary to meet the needs of the tumor cell such as uncontrolled proliferation, dissemination, and acquisition of resistance to death processes induced by both unfavorable environmental conditions and therapeutic drugs. Many oncogenes and tumor suppressor genes have a significant effect on cellular metabolism, as there is a close relationship between the pathways activated by these genes and the various metabolic options. The metabolic adaptations observed in cancer cells not only promote their proliferation and invasion, but also their survival by inducing intrinsic and acquired resistance to various anticancer agents and to various forms of cell death, such as apoptosis, necroptosis, autophagy, and ferroptosis. In this review we analyze the main metabolic differences between cancer and non-cancer cells and how these can affect the various cell death pathways, effectively determining the susceptibility of cancer cells to therapy-induced death. Targeting the metabolic peculiarities of cancer could represent in the near future an innovative therapeutic strategy for the treatment of those tumors whose metabolic characteristics are known.

## 1. Introduction

In cancer cells, profound metabolic changes are often observed compared to untransformed cells that affect numerous functions ranging from the absorption of nutrients to the biosynthesis of macromolecules, to the production of energy. In fact, all the main metabolic pathways appear altered in tumor cells: glycolysis, tricarboxylic acid (TCA) cycle, glutaminolysis, oxidative phosphorylation (OXPHOS), and pentose phosphate pathway (PPP). Many oncogenes and tumor suppressor genes have a significant effect on cell metabolism. Indeed, there seems to be a close relationship between the pathways activated by these genes and the metabolic options necessary to meet the needs of the cancer cell: uncontrolled proliferation, dissemination (invasiveness and metastasis), and acquisition of resistance to both physiological and drug-induced cell death processes. These metabolic peculiarities, and the underlying molecular pathways, could therefore represent specific therapeutic targets for the treatment of some tumors whose metabolic characteristics are known. This could represent a first step towards a personalized therapy designed on the biological specificities of a particular tumor.

At the base of the development of a tumor there are genetic and epigenetic modifications that accumulate over time causing alteration of the proliferation giving the cell a potential immortality. Although the genetic and epigenetic mutations observed in different tumors may be different, they very often lead to the alteration of a small number of signaling pathways. Among these, AKT/PI3K, MYC, AMPK, RAS-ERK, P53, mTORC, and Wnt/β-catenin appear to be those most involved in tumorigenesis. Interestingly, all these influence cell metabolism [1].

## 2. Mitochondria and Cancer Cell Metabolism

Mitochondria regulate cell metabolism and energy production through the OXPHOS process, which also represents the most important source of reactive oxygen species (ROS) production, generating almost 90% of the total cellular ROS [2] (the main intracellular sources of ROS are mentioned in Figure 1). ROS are molecules containing oxygen derived from incomplete reduction of O_2_, and include superoxide (O_2_^•−^), hydrogen peroxide (H_2_O_2_), and hydroxyl radical (OH^•^).

The OXPHOS process produces ATP from ADP and inorganic phosphate by transport of electrons to a series of transmembrane protein complexes located in the mitochondrial inner membrane [3]. The OXPHOS system comprises five enzymatic complexes: complex I (NADH-ubiquinone oxidoreductase), complex II (succinate-ubiquinone oxidoreductase), complex III (ubiquinone-cytochrome c oxidoreductase), complex IV (cytochrome c oxidase), and complex V (ATP synthase). Altogether, the respiratory complexes form the electron transport chain (ETC), which creates an electrochemical gradient to produce ATP. Respiration, due to redox nature of its reactions, inevitably produces ROS as a by-product [4]. In fact, the loss of electrons along the respiratory chain represents the main cause of mitochondrial ROS accumulation [5].

Complex I is often mutated in cancer where it represents a major source of ROS. In the complex III of the ETC, it is mainly the Q-cycle that determines the production of O_2_^•−^. This occurs when complex III transfers electrons from complexes I and II (respectively) to cytochrome c [6]. Interestingly, overexpression of complex III subunits has been observed to induce tumorigenesis in a ROS-dependent manner in colorectal cancer [7], lung cancer, and hepatocarcinoma [8].

Recent literature data highlight that mitochondrial metabolism can be considered a target for anticancer therapy, with a particular focus on OXPHOS inhibition [9,10,11]. Lim and coworkers identified three different metabolic pathways capable of guaranteeing the survival of cancer cells: OXPHOS, glutamine, and glucose metabolic pathways. Therefore, inhibiting only one of these pathways may not be the most appropriate anticancer strategy [12]. In particular, in melanoma they demonstrated the existence of a subset of cells that used mostly OXPHOS and, through the overexpression of peroxisome proliferator-activated receptor gamma coactivator 1-α (PGC1α), were resistant to oxidative stress. Inhibition of PGC1α led to an increase in ROS production, HIF1α stabilization, and a metabolic shift towards glycolysis. Subsequent inhibition of glycolysis with pharmacological treatments could induce a further metabolic switch towards the use of glutamine as an energy source. Thus, the existence of three alternative metabolic options for cancer cells underlines the importance of a combined therapeutic strategy [12]. Hepatocellular carcinoma and ovarian, pancreatic, colon, and breast cancers emerged as tumors relying on OXPHOS [12,13,14,15,16]. In particular, in hepatocellular carcinoma it has been demonstrated that the growth and survival of the tumor itself are decreased by using antiparasitic drug Atovaquone that causes oxidative stress and oxidative DNA damage.

The mitochondria ROS production must be constantly balanced to avoid damage to proteins, mitochondrial DNA, and membrane lipids. For this purpose, mitochondria have two main antioxidant systems: the superoxide dismutase, a manganese-requiring mitochondrial enzyme (MnSOD), and the Thioredoxin (Trx) system. Mitochondrial redox systems are strictly involved in cancer favoring cell growth and preventing apoptosis. In accordance with this, in drug-resistant cancers Trx system was found upregulated and represented a negative prognostic factor in several types of cancers [17]. Indeed, cancer cells often shown high levels of ROS. In particular, it was shown that Trx reductase, influencing cell proliferation, invasion, and migration, has an important role in colon and rectal cancer, prostate cancer, hepatocarcinoma, osteosarcoma, as well as in the progression of lung cancer [17]. An experimental study conducted by Li and co-workers on diffuse large B cell lymphoma cell lines demonstrated that the overexpression of Trx was associated with cell growth and survival, as well as with the development of chemoresistance [18]. Chronic and uncontrolled oxidative stress is involved in the onset of numerous human pathologies, including cancer (Figure 2). In general, in tumors the elevated ROS amount induces the upregulation of the antioxidant pathways. In particular, ROS production is regulated through nuclear factor (erythroid-derived 2)-like 2 (NRF2), the main factor responsible for the cellular antioxidant response. As mentioned above, NRF2 have also a role in metabolic reprogramming of cancer cells by regulating some enzymes involved in the PPP and glutamine metabolism [19]. The PPP is a parallel pathway to glycolysis used to synthesize nucleotides and to produce NADPH molecules necessary for the regeneration of GSH [20]. The amino acid glutamine, in addition to its role in protein synthesis, can be used as a substrate for the TCA cycle becoming the main energy source in place of glucose for cancer cells. Inhibition of glutamine metabolism by bis-2-(5-phenylacetamido-1,3,4-thiadiazol-2-ylethyl sulfide (BPTES) has been observed to increase ROS production resulting in growth inhibition of several cancers [21,22]. In accordance with this, glutaminase inhibitors have been proposed as potential anticancer agents [23].

Given the essential role played by mitochondria in cancer, many drugs or natural compounds perform their antitumor activity by targeting certain mitochondrial functions, such as OXPHOS, and glutamine metabolism. Some of these drugs show a synergistic action when combined together because of their ability to prevent the metabolic switch from OXPHOS to glycolysis, able to support the energy needs to cancer cells to favor their own survival following anticancer treatment. For example, the combined use of two non-steroidal anti-inflammatory drugs, Diclofenac and Lumiracoxib, has been shown to increase the action of the RAF inhibitor Vemurafenib in melanoma cells [24]. In particular, Diclofenac and Lumiracoxib, decreasing lactate secretion and affecting oxidative phosphorylation, when combined with Vemurafenib increased the anti-glycolytic impact of the RAF inhibitor and prevented the metabolic reprogramming of cancer cells to OXPHOS [24]. A similar effect was observed in breast cancer cells when the treatment with a Blc-2 inhibitor, Venetoclax or WEHI-539, was performed in combination with the glycolysis inhibitor 2-Deoxy-D-glucose (2DG).

Among many functions, mitochondria also regulate apoptosis, by releasing pro-apoptotic factors and controlling calcium homeostasis.

Since resistance to apoptosis is one of the hallmarks of cancer, in the last few years many new compounds have been conceived to overcome apoptosis resistance [25]. Among them, Raptinal is an unusually rapid inducer of caspase-dependent apoptosis in multiple cancer cell lines and an in vivo cancer model [26] able to induce tumor growth inhibition and apoptosis via oxidative stress within hours of administration in hepatocellular carcinoma [27].

Mitochondria constitute a network in continuous reorganization through the processes of fusion/fission and their redistribution within the cell [28]. The balance between fission and fusion determines the shape, size, and number of mitochondria, strongly impact on energy metabolism. Alterations of this balance contribute in various ways to tumorigenesis and tumor progression [29].

Mitochondrial dynamics profoundly influence also cell metabolism. Indeed, many observations suggest that mitochondrial elongation correlates with high OXPHOS activity and ATP production, consistent with the hypothesis that elongated mitochondrial networks are more efficient in energy generation [30,31]. On the contrary, the fission and consequent redistribution of mitochondria to the leading edge promotes tumor invasion by providing ATP and metabolic intermediates necessary for the biosynthetic demand of the cancer cell. Equally important is the transfer of functional mitochondria from healthy cells, especially the stromal ones, to tumor cells via nanotubular structures, which contributes to the metabolic plasticity of cancer thus promoting cell proliferation and survival in prohibitive microenvironmental conditions [32,33]. In light of this, mitochondrial dynamics could represent an innovative therapeutic target and/or a useful prognostic biomarker in cancer.

## 3. Glucose Metabolism Alteration in Cancer Cells

Glycolysis is the main metabolic pathway that allows mammalian cells to produce ATP. Glucose is transported into cancer cells via different glucose transporters (GLUT), in particular by GLUT1 [34], that are overexpressed in many types of tumor cells [35]. In the body, excess glucose is stored as glycogen, mostly in specialized organs like the liver or muscle, but also in other peripheral tissues [36]. When glucose concentration is reduced, glycogen can be mobilized, releasing glucose-1-phosphate (G-1P), which is converted to glucose-6-phosphate (G-6P) and utilized by glycolysis [37]. In healthy cells, after glucose uptake the enzyme hexokinase (HK) phosphorylates intracellular glucose to form G-6P. In mammals, five HK isoenzymes (HK1-4 and HKDC1) have been characterized of which only HK2 has two functional enzymatic pockets. HK1 is ubiquitous and represents the most abundant form in most tissues. HK2, although less expressed than HK1, is the main isoenzyme in insulin-sensitive areas such as heart, skeletal muscle, and adipose tissue, and in a wide range of tumors [38,39]. Subsequently, after the isomerization of G-6P into fructose-6-phosphate (F-6P), phosphofructokinase (PFK) catalyzes the phosphorylation of F-6P into F-1,6-2P that is converted to glyceraldehyde-3-phosphate (GA-3P) by the enzyme aldolase. This latter enzyme is also responsible for the production of dihydroxyacetone phosphate (DHAP) that, in turn, produces other molecules of GA-3P [40]. The GA-3P is converted into 1,3-bisphosphoglycerate (GA-2P) by glyceraldehyde-3-phosphate-dehydrogenase (GAPDH), and, subsequently, phosphoglycerate mutase 1 (PGAM1) catalyzes the reversible conversion of 3-phosphoglycerate (3P-G) and 2-phosphoglycerate (2P-G). The enolase converts 2P-G into phosphoenolpyruvate (PEP), which is finally and irreversibly converted to pyruvate by the enzyme pyruvate kinase (PK) getting ATP and NADH. 

PK is one of the key enzymes in glycolysis. It encompasses four different subtypes (PKM1, PKM2, PKL, and PKR), encoded by two sets of genes [41]. All have a tetrameric form and pyruvate kinase activity, but only PKM2 has both a dimeric and a tetrameric form. The expression of PKM1 and PKM2 is also regulated by the c-myc oncogene, which promotes the expression of PKM2 [42]. Unlike PKM1, PKM2 has been reported to promote the conversion of glucose to lactate [43]. According to this, PKM2 is mainly expressed during embryonic development, is closely related to tissue regeneration, and plays an important role in tumors. In its dimeric form it acts as a transcription factor and participates in the epithelial–mesenchyme transition (EMT), and in the processes of invasion and metastasis [44]. Accordingly, PKM2 is found overexpressed in different cancer cell types [45,46]. It has also been hypothesized that the dimer/tetramer ratio of PKM2 determines whether glucose metabolism leads to the biosynthesis of nucleic acids, proteins, and amino acids or to energy metabolism [47].

Finally, the pyruvate generated by PK is converted into acetyl-CoA, synthesizing ATP and CO_2_ through OXPHOS. In the presence of limited amounts of oxygen, glucose is transformed in lactate (anaerobic glycolysis). In fact, under hypoxic conditions, the pyruvate produced by glycolysis is converted into lactate by the lactate dehydrogenase (LDH) (Figure 3). Glycolysis can also occur in the presence of sufficient amount of oxygen (Warburg effect, see below). In this case, we refer to aerobic glycolysis that generally implicates the repression of respiration [48]. Anyway, all metabolic options are often active in cancer cells where glucose uptake increases dramatically to meet their high-energy demand [34].

Remarkably, tumor cells can activate both the Warburg effect, thus synthesizing large amounts of lactate regardless of the availability of oxygen, and OXPHOS simultaneously [34,37]. Moreover, in malignant tumor cells several glycolytic enzymes can be upregulated. For example, the PGAM1 [49], PKM2 [50], HK2 [51], and GAPDH [52] are often upregulated in tumor cells, PFK results increased in various breast tumors [53], enolase in pancreatic ductal adenocarcinoma [54], and aldolase in the lung’s squamous cell carcinoma [55]. In summary, all these enzymes may be potential targets for the treatment of malignant tumors. Additionally, LDH could be utilized as a therapeutic target. In fact, LDH inhibition was observed to suppress the progression of lymphomas and pancreatic cancer in vivo [56].

## 4. Warburg Effect and Glucose Metabolism Reprogramming

Otto Warburg in 1925 observed that most cancer cells were characterized by an altered glucose metabolism [57,58,59]. Multiple studies have reported that in non-cancer cells ATP was usually generated via OXPHOS, which led to the actual production of 30–32 molecules of ATP per molecule of oxidized glucose, therefore, less than theoretically expected of 36–38 molecules [59,60,61]. Normally, cells internalize glucose through glucose transporters, and metabolize it to pyruvate in the cytosol by the glycolytic process. Under physiological conditions, the pyruvate is oxidized in the mitochondrial matrix to acetyl coenzyme A (CoA), which is used to generate with high efficiency ATP through OXPHOS [61]. In conditions of poor oxygenation, normal cells shift versus the alternative, less efficient, glycolytic pathway, the so-called Embden–Meyerhoff fermentative pathway. In this case, one molecule of glucose is converted into 2 molecules of lactic acid and 2 H^+^ to produce only 2 molecules of ATP. Thus, in normal cells, the balance between OXPHOS and glycolysis is determined by the amount of oxygen available [62].

Otto Warburg had observed that cancer cells preferentially used the less efficient glycolytic (fermentative) pathway to produce energy even in the presence of adequate levels of oxygen (aerobic glycolysis) [57]. Further findings reported that glycolysis, although less efficient is more rapid than respiration in producing ATP [59,63]. In fact, it has been reported that 24 molecules of ATP are produced through aerobic glycolysis compared to 1 molecule of ATP produced by respiration in the same unit of time [64]. For several years, the molecular and biological mechanisms of the Warburg effect remained unclear and quite unexplored. With the advent of FdG-PET imaging technique the interest in understanding the molecular mechanisms underlying metabolic reprogramming observed in cancer cells received a renewed interest, and the Warburg effect has been studied extensively. These studies provided insights into how the aerobic glycolysis emerges in tumor cells, confirming that many cancers upregulate glycolysis, and take up much more glucose than normal cells [65]. Although Warburg initially attributed the observed altered glucose metabolism to inefficient mitochondrial respiration [66], further data reported that most of the cancers have fully functioning mitochondria, and only few of them present mitochondrial dysfunctions [67], supporting the hypothesis that cancer cells use the aerobic glycolysis even in the presence of functional mitochondria. Crabtree and colleagues first demonstrated that cancer cells exhibited variable levels of mitochondrial respiration [61,68], which was under the regulation of glucose that works in a feedback loop inhibiting respiration therefore suppressing the mitochondrial oxidation of pyruvate [59,68]. Further studies confirmed the observations of Crabtree, leading to the notion that even if cancer cells can still exhibit respiration, glycolysis is coupled with lactate production and uncoupled with oxidative metabolism of glucose. Therefore, the glucose is directed versus the production of lactate and the biosynthesis of macromolecules in most of cancers. Moreover, additional studies have reported that mutations in the mitochondrial DNA, or a decrease in the number of functional mitochondria could contribute to the Warburg effect, promoting tumor growth and progression, multidrug resistance, and metastasis formation [69,70,71,72]. 

## 5. Tumor Microenvironment and Cancer Cell Metabolism

Besides genetic and epigenetic changes known for a long time to be involved in tumor formation and development, a new aspect of carcinogenesis has been the focus of intense studies in the last two decades from oncology. Indeed, the interest in studying the tumor microenvironment (TME) and its potential role in driving the selection versus the cancerous phenotype was growing [60,73]. From these studies, it emerged the awareness that TME plays a fundamental role in the development of the tumor and in its progression towards a malignant phenotype. Moreover, it has also been demonstrated that the interplay between tumor cells and their surrounding microenvironment represents an important determinant in tumor response to therapies (Figure 4 schematically shows the cross talk between tumor cells and TME). The TME is characterized by cellular components represented by proliferating tumor cells and tumor stromal cells (i.e., stromal fibroblasts, endothelial cells, immune cells, and infiltrating inflammatory cells). It also presents non-cellular components of the extracellular matrix such as collagen, fibronectin, hyaluronan, laminin, and ecc. [74,75]. Within the TME, tumor and non-tumor cells are kept in touch through secreted soluble molecules, typically metabolites, cytokines, chemokines, microRNA, growth factors, inflammatory mediators, and extracellular matrix remodeling enzymes, as well as through the recently emerging circulating tumor cells (CTCs), exosomes, cell-free DNA (cfDNA), and apoptotic bodies [76,77]. The interactions between cellular and non-cellular components of the TME, through the above mediators, activate several signaling pathways, which can act by favoring or disadvantaging tumor progression, invasion, and drug resistance, as well as contributing to cancer-associated angiogenesis [78,79]. Often the tumor environment is characterized by hypoxia, lack of nutrients and acidity, non-permissive conditions for non-tumor cells that drive clonal selection towards highly invasive and metastatic cell phenotypes. In order to survive in this hostile microenvironment, and to sustain their high proliferation rate, many solid cancers evolve versus the use of glycolysis to generate ATP even in condition of adequate oxygen concentrations (Warburg effect) [60,65,80]. The production of ATP through aerobic glycolysis is significantly lower than that produced by OXPHOS and it is associated with a high production of lactate. To meet their energy needs, the cancer cells therefore increase their glucose intake and proton extrusion, thus contributing to the acidification of TME, which directly regulates tumor invasion and metastasis [60,65], as well as drug resistance [81].

### 5.1. Hypoxia and Acidic pH of TME

Hypoxia is a well-characterized and common trait of the TME, which occurs early during tumor development. It is caused by an inadequate oxygen supply at the tumor core level as a consequence of an insufficient microvasculature. Cancer cells are usually highly proliferative and need a huge amount of nutrients and oxygen in order to compensate their rapid turnover. The high rate of proliferation, generating an increase in the tumor mass, leads to a distancing of the tumor core from the blood vessels, which causes hypoxia and consequent cell necrosis. Solid tumors are now believed to be heterogeneous in oxygenation, presenting hypoxic regions and well oxygenated areas that mutually influence each other from a metabolic point of view. Hypoxic areas usually are distant from blood vessels, and consequently from oxygen. In order to survive in this hostile environment, hypoxic cancer cells need to adapt to the low-oxygen conditions, which leads to the selection of phenotypes more aggressive, resistant to apoptosis triggering signals, and to a variety of commonly used chemotherapeutic agents and radiations [83]. A key actor of this adaptation is HIF and its signaling pathways [84]. HIF is a heterodimeric transcription factor composed of a constitutively expressed β-subunit and an O_2_-regulated α-subunit [85]. Mammals present three HIF-α isoforms (HIF-1α, HIF-2α, and HIF-3α) and three HIF-β isoforms (ARNT/HIF-1β, ARNT2, and ARNT3) each encoded by a different gene [86]. HIF-1α and HIF-1β are expressed ubiquitously in human and mouse, whereas all the other isoforms are tissue specific. HIF-1α normally has a very short lifespan. Indeed, in presence of oxygen, it is hydroxylated and enters into the ubiquitin-proteasome-system where it is degraded [87]. When oxygen is scarce, hydroxylation is inhibited and HIF-1α is stabilized, translocates into the nucleus, and heterodimerizes with a HIF-1β subunit to form a HIF complex. This complex binds to the hypoxia response element (HRE) in the promoter and enhancer region of the target genes and is responsible for tumor cell survival, glycolysis increase, angiogenesis, and invasion/metastasis. Growth factors, cytokines, and reactive oxygen and nitrogen species are able to activate HIF-α [88].

Due to their ability to directly activate transcription of target genes, an increased expression of HIF-1α and HIF-2α was reported in a variety of human cancers and was correlated with a switch versus a more aggressive phenotype [89]. Generally, both HIF-1α and HIF-2α show a positive correlation with tumor progression, but with some exceptions. In a xenograft model of colorectal carcinoma HIF-1α knock down by a small interfering RNA was reported to impair tumor growth, while knocking down HIF-2α led to an increase [90]. In the same vein, HIF-1α overexpression was observed to slow down the growth of renal carcinoma in xenografts, whilst an increased HIF-2α expression favored the tumor growth [91]. Furthermore, either in head and neck cancer or in non-small-cell lung patients the HIF-1α overexpression was associated with a reduction in the mortality rate, while the reverse was reported for HIF-2α [92,93,94].

Interestingly, HIF-1α could regulate glycogen metabolism under hypoxic condition and nutritional deficiencies promoting accumulation of glycogen [95,96]. For this reason, inhibition of HIF-1α, or HIF-1α-mediated metabolism pathway, may represent a potential treatment for cancer [97,98]. Indeed, it has been observed that many drugs, including Aspirin [99], Dovitinib [100], Metformin [101], Cetuximab [102], and Tamoxifen [103] could block tumor growth by inhibiting, directly or indirectly, the expression or activity of HIF-1α.

Together with the so-called “Warburg effect” and hypoxia, acidosis of the TME occurs early during the metabolic switch of cancer cells towards anabolic processes. Several studies have related tumor extracellular acidosis to invasiveness, metastatic behavior, and resistance to cytotoxic agents [104,105,106,107], and therefore acidosis is nowadays considered a hallmark of malignancy. The first evidence showing that most human cancers are acidic goes back a long time basing on microelectrode measurements of tumor pH [108]. With the advent of magnetic resonance spectroscopy (MSR), a technique able to measure the tumor pH more precisely and accurately than microelectrodes, researchers demonstrated that the pH of human tumors is neutral-alkaline (7.0–7.4) inside the cells, while the extracellular pH of the same solid tumors is acidic (6.0–6.9). This indicates that the pH gradient of cancer cells is acidic outside [109]. Hence, differently from normal cells presenting a pH gradient that is alkaline outside, the pH gradient across the plasma membrane of many human cancer cells is acid outside and then is reversed compared to that of normal cells. 

Mounting data supported the hypothesis that the establishment of an acidic environment surrounding tumor cells occurs early during the selection against highly malignant phenotypes because of the hostile conditions unsuitable for life of most cells. These studies also highlighted that the acid TME is the consequence of the tumors switch versus a high proliferation rate and an aberrant metabolic phenotype. These features, together with low nutrient and oxygen levels typically present in the TME, favor the selection of cells able to survive in these adverse conditions. This implies enormous glucose utilization and amino acid metabolism, sugar fermentation, and release of lactate, which in turn induces release of large amount of protons into the extracellular TME [110], contributing to the acidification of TME [111]. Tumor extracellular acidosis is also induced by the production of carbonic dioxide through mitochondrial respiration of cancer cells, leading to the release of an additional amount of protons into the TME [112,113].

Therefore, in order to survive in these extremely toxic conditions and to evade acid-mediated toxicity, cancer cells upregulate the expression and activity of several proton extrusion mechanisms [114] able to release excesses protons and lactate into extracellular environment, avoiding the acidification of the cytosol. Among proton flux regulators are vacuolar H+-ATPases (V-ATPases) [114,115,116], Na+/H+ exchanger (NHE), monocarboxylate transporters (MCTs), carbonic anhydrase IX/XII (CA-IX/XII) [117,118], and Na+/HCO3 co-transporters (NBCs) [119]. Literature data on different human cancer cell types correlated the increased expression and activity of these proton exchangers with tumorigenesis, metastasis, invasiveness, resistance to apoptosis, and multidrug resistance [120,121,122,123]. Thus, a strategy aimed at inhibiting their expression and activity was considered by researchers, an appealing approach in order to switch on cancer cell death and therefore to provide new insights and alternative strategies for treating cancers. For this reason, several inhibitors with a great anticancer potential have been identified and developed and are under exploitation for the setting up of novel and promising antitumor strategies. Among the V-ATPase inhibitors, the anti acid compounds Proton Pumps Inhibitors (PPIs) were extensively investigated in preclinical and clinical studies in a wide range of tumors. Literature data suggested their potential use either as adjuvants in chemotherapeutic treatments, and as novel and alternative antitumor drugs, with less toxicity for tumor patients and more specificity against the acidic environment of tumors [105,124,125,126,127,128,129,130,131,132,133,134]. Evidence about NHE1 inhibitors reported antitumor and chemo-sensitizing effects in different human tumors [135,136,137,138]. Evidence showed that inhibition of NHE1 using specific molecules promoted intracytosolic acidification in cancer cells, strongly favoring reduction in cell proliferation and activation of apoptosis in different types of cancer [139,140]. Differently to the others, the amiloride derivative HMA triggers cancer cell apoptosis in a caspase-independent manner, involving autophagy [141,142,143], or programmed necrosis in breast cancers [144]. Similar results were found about MCT inhibitors [145,146]. Additional very promising anticancer agents are pharmacological inhibitors targeting specifically CA-IX and –XII [118,147,148,149]. Some of these drugs showed a greater action when combined together. 

Studies conducted so far highlighted that tumor hypoxia, tumor extracellular acidity, and tumor glycolytic metabolism are important determinants of resistance to classical chemotherapeutic agents, of innate as well as acquired therapy-induced resistance. Literature data evidenced that the acidic TME impaired anticancer drugs distribution, by inducing protonation of many chemotherapeutic drugs, leading to their neutralization and inactivation. Thus, inactive drugs cannot pass through the plasma membrane or are sequestered into intracellular acidic vesicles or endosomes. As a result, the drug intake, and consequently its cytotoxic effect, is significantly reduced. Tumor cells became then resistant to cytotoxic agents. Moreover, evidence underlines that HIF-1α can induce the expression of the genes encoding for the drug efflux transporters, which contains into their promoters hypoxia responsive elements [150]. In addition, studies performed in various tumor cell lines observed that HIF-1α regulated the expression of the drug efflux transporters that extrude different chemotherapeutic drugs from the cytosol to the extracellular compartments favoring in turn a decrease of the drug intracellular concentration. HIF-1α can also play an inhibitory role against apoptosis, the cell death usually occurring following treatment with most chemotherapeutics.

### 5.2. Lactate Shuttle and the Reverse Warburg Effect

The Warburg effect is the prevailing theory explaining the metabolic reprogramming observed in cancer cells and is considered a hallmark of cancer, but it does not provide a complete explanation of the aspects underlying the tumor metabolism and the molecular mechanisms that underlie it are not yet completely understood. Recent evidence highlighted that the majority of cancer cells contain mitochondria fully functioning and still able to utilize OXPHOS, and that there is a metabolic intratumoral heterogeneity [11,151,152]. This means that different metabolic phenotypes coexist within the tumor mass, and then some cancer cells produce energy through aerobic glycolysis while others predominantly via mitochondrial OXPHOS [145,153]. Tumor heterogeneity depends on the tumor oxygenation, which fluctuates from normoxia to hypoxia [154,155]. Therefore, within the tumor mass there are areas well oxygenated, whereas others are hypoxic. This feature is strongly dependent on the neovasculature induced by cancers [154,155]. Furthermore, a recent theory was introduced to explain tumor heterogeneity regarding glucose metabolism. 

In 2009, a renewed and modified version of the “Warburg effect” theory, called the “Reverse Warburg effect” [155,156,157,158], emphasized the peculiar role played by the cells of the tumor stroma in metabolically supporting cancer cells favoring their proliferation and growth, and described the existence of a close interplay between cancer cells and stromal cells, in particular cancer-associated fibroblasts (CAFs). Basing on this theory, hypoxic cancer cells induce the nearby CAFs to follow aerobic glycolysis and then to produce lactate that is released in the TME through MTC1–4 transporters. Lactate is then transferred back to oxygenated tumor cells (or oxygenated CAFs) where it is utilized by OXPHOS to produce further ATP and metabolites necessary for cancer cell survival [62,145,155,157,158]. The remaining lactate is stored in the extracellular TME contributing to acidification. The tight metabolical interplay between tumor cells and CAFs would support the metabolite exchange and provide cancer cells not only of ATP, but also of other molecules necessary for their proliferation and growth, and useful to limit cell death [62,153,159].

Therefore, the lactate produced by cancer cells contributes to their proliferation, to the acquisition of a phenotype resistant both to drug-induced apoptosis [160] and to anoikis, thus also favoring metastatic dissemination [161].

In this context, lactate cannot be considered a simple waste product of glycolysis, but rather a messenger capable of determining an adaptive metabolic reprogramming that contributes to tumor progression. 

Key players of this metabolic cross talk are the lactate transporters MCT1 and MCT4 [62,155,159]. In particular, MCT1 is involved in the intake of lactate in oxidative active cancer cells, while MCT4 transports lactate out of glycolytic cells [145]. According to this, it was reported the overexpression of MCT1 in oxygenated cells, and of MCT4 in hypoxic cells [145,162]. Interestingly, in co-culture experiments, Whitaker and coworkers demonstrated that MCT4 is selectively expressed by CAFs only when co-cultured with breast cancer cells [156]. Moreover, studies aimed at inhibiting MCT1 reported a glycolytic reprogramming of cells, whereas inhibition of MCT4 leads to increased intracellular levels of lactate, in turn favoring cell death [59,145,155]. Findings also suggested that shuttling of lactate would prevent the formation of an extremely acidic TME, probably even for the cancer cells [163,164]. In addition, extracellular accumulation of lactate induced by glycolytic hypoxic tumor cells was reported to stimulate angiogenesis and tumor growth through a pathway mediated by IL-8 [59,155,165].

The metabolic cross talk between CAFs and cancer cells involves, among others, Caveolin-1 (CAV1), HIF-1α, nuclear factor kappa-light-chain-enhancer of activated B cells (NF-κB), and TP53-induced glycolysis and apoptosis regulator (TIGAR). It has been observed that the lack of CAV1 expression in CAFs determined a mitochondrial dysfunction, orienting their metabolism towards glycolysis [157,158]. Interestingly, the transcription factors HIF-1α and NF-κB were correlated with decrease of CAV1 in CAFs [166]. In this regard, it was suggested that tumor cells by promoting oxidative stress in CAFs could activate HIF-1α, which in turn promoted autophagy and glycolysis through the hypoxia-mediated degradation of CAV1 and enhanced either glucose transporters or glycolytic enzymes [166,167]. Additionally, activation of NF-κB in CAFs was reported to induce similar effects [166]. As far as TIGAR is concerned, it plays a key role in supporting the “reverse Warburg effect”. In fact, in breast carcinoma cells overexpressing TIGAR, a glycolytic inhibitor under the control of p53, a decreased glycolysis, and a concomitant increased OXPHOS were reported [159,168]. Interestingly, when breast carcinoma cells overexpressing TIGAR were co-cultured with CAFs, these latter shifted versus a glycolytic phenotype [159,168]. High levels of TIGAR were found in different types of tumors [169,170,171] and mutated forms of p53 were observed to promote the expression of TIGAR [172].

## 6. Cell Death and Metabolism in Cancer

According to the Nomenclature Committee on Cell Death, cell death can be distinguished in an accidental and in a regulated form, the latter triggered and controlled by specific biochemical pathways [173]. The fine knowledge of the molecular mechanisms involved in the different types of regulated cell death makes their pharmacological and genetic regulation possible, at least potentially.

Cell death mechanisms are often compromised in cancer cells. Loss of sensitivity to the cell death signals and immortality is one of the major characteristics of cancer cells. The metabolic adaptations observed in cancer cells not only favor their proliferation and invasion, but also their survival by inducing intrinsic and acquired resistance to different antitumor agents [174,175]. Indeed, the literature extensively reported that the resistance to the conventional cell death pathways represents an additional hallmark of cancer. Among them, apoptosis is mainly responsible for cell death following treatment with most anticancer agents. Autophagy, and the more recently discovered ferroptosis and necroptosis, also play a peculiar role.

### 6.1. Apoptosis

Apoptosis is a type of programmed cell death characterized by membrane shrinkage, chromatin condensation, nuclear fragmentation, plasma membrane blebbing, and release of small vesicles (apoptotic bodies) into microenvironment, which are engulfed by phagocytes and neighboring cells. This process is mediated by a family of cysteine aspartyl proteases, known as caspases that, once activated by proteolytic cleavage, activate the effector caspases, leading to cell death execution [176,177,178].

Apoptosis can involve two different pathways: an intrinsic mitochondrial pathway, activated following the alteration of cellular homeostasis, such as limiting levels of growth factors, oxidative stress, and DNA damage, and an extrinsic receptor-mediated pathway, which involves the activation of specific death receptors [178,179]. In particular, the extrinsic pathway is triggered by ligand/receptor interaction between a tumor necrosis factor (TNF) family member and the corresponding transmembrane death receptors. Once binding occurs, adapter proteins are required to trigger the execution phase of apoptosis [180]. Both pathways converge at mitochondria inducing the permeabilization of the outer mitochondrial membrane (MOMP) [181]. For more details, see Figure 5.

The most common mechanism used by cancer cells to escape apoptosis, thereby gaining survival advantages, is represented by the upregulation of anti-apoptotic proteins, such as Bcl-2, Bcl-XL [182,183]. This implies that a greater amount of the pro-apoptotic proteins (i.e., Bax and/or Bak) are deactivated, conferring resistance to apoptosis [184].

Literature data reported that cancer cells harness altered metabolism to evade apoptosis. Indeed, it has been observed that under hypoxic conditions, HIF-1 induced the upregulation of glucose uptake and glycolysis, leading to resistance to apoptosis [185,186,187,188].

Recent studies have suggested that silencing HK2, a glycolytic enzyme often upregulated in cancer cells, was associated with increased apoptosis in prostate cancer models [189]. In fact, HK2 would prevent the release of cytochrome c, therefore the occurrence of apoptosis, inhibiting the insertion of Bax into the mitochondrial membrane and its dimerization with Bak [190,191].

P53 is a tumor suppressor gene that, in conditions of stress, can induce the arrest of cell growth in order to repair the cell damage, or trigger apoptosis if the damage is not repairable [192]. This gene is found mutated in about 50% of human cancers and most mutations compromise its activity [193]. Interestingly, p53 is modulated by changes in glucose metabolism. Once cellular glucose levels are low, p53 becomes active and promotes cell growth arrest [194]. P53 in turn blocks glycolysis, downregulating GLUT1 and GLUT4. Therefore, p53 mutations were able to enhance the cellular levels of glucose. At the same time, p53 can increase glycolysis induced by the TP53-Induced Glycolysis and TIGAR, which stop the action of phosphofructokinase (PFK1) [184].

An additional key anti-apoptotic actor is the serine/threonine kinase, AKT, which is mostly active in cancer cells, favoring their survival by promoting glycolysis and lactate production [175,195,196]. It has been reported that AKT prevented apoptosis at different levels. For instance, in the presence of glucose, AKT negatively regulates the induction of PUMA [175,197,198] and suppresses the degradation of MCL-1, favoring its neo synthesis [198,199]. Moreover, AKT was also able to induce the translocation of GLUT1 to the plasma membrane [196], and the mitochondrial localization of HK2 [200,201], preventing the release of cytochrome c [201,202].

In recent years, many efforts have been made to identify new multi-targeting agents and devise therapies capable of targeting cancer cells. Among these, polyphenols, particularly Resveratrol, represent an excellent candidate as multi-targeted drugs to supplement chemotherapy. Resveratrol influences several vital stages of tumors in a wide range of cancer types. Indeed, it inhibits proliferation and migration and induces apoptosis by modulating signal transduction cascades and glucose metabolism in various cancer types, including breast, lung, colorectal, prostate, ovarian, leukemia, liver, and pancreatic cancers [203,204,205,206].

Moreover, targeting tumor metabolism or the TME of solid tumors are under exploitation in order to restore tumor sensitivity to apoptosis induced by anticancer drugs. For example, approaches aimed at targeting tumor acidity through inhibition of proton pumps were investigated. Studies reported that pharmacological inhibition of V-ATPase with bafilomycins and concanamycins, demonstrated efficacy in inducing apoptosis in human cancer cells and in inhibiting their growth [207,208,209]. Unfortunately, these drugs were highly toxic for normal cells, limiting their potential application in clinical settings. Other V-ATPase inhibitors that have been investigated and suggested as possible new anticancer drugs were the anti-acid compounds, PPIs (Proton Pumps Inhibitors), currently used to treat peptic diseases. Two peculiar PPIs properties rendered these compounds very appealing, tumor selective and acidity targeting drugs: their activation in acidic environment, for instance that of tumors, and the absence of systemic cytotoxicity for normal cells [210]. Preclinical data together with clinical studies suggested that these compounds, by modulating tumor acidity, can induce apoptosis in a caspase-dependent or independent manner, restoring sensitivity to many chemotherapeutic drugs, and exerting per se a strong cytotoxic activity against a wide range of tumor cell lines [105,130,131].

Natural compounds capable of targeting enzymes of the glycolytic pathway are also under investigation. For instance, findings suggested that the natural compound lapachol inhibited PKM2, inducing a blockade of glycolysis, decreasing ATP levels and sensitizing melanoma cells to apoptosis [211]. In addition, in prostate cancers lapachol promotes apoptosis by inducing increase in ROS production and in the transcription of caspase-3 and -9 [212]. Apoptosis induction by lapachol was also observed in tumors of different histology [213,214].

Lectins, in particular Polygonatum cyrtonema lectins (PCL), were also able to trigger apoptosis in prostate cancer cells targeting HK2, leading to decreased glucose consumption and lactate production [215]. This ability of PCL passes through inhibition of the PI3K/Akt pathway, probably induced by competition in EGFR binding [215]. In hepatocellular carcinoma, triggering of apoptosis by compound K, a metabolite of the ginsenosides, was induced by inhibition of glycolysis via blockade of AKT/mTOR/c-Myc pathway, HK2 and PKM2 [216].

Literature data suggested that the tumor metabolic reprogramming sensitizes cancer cells to non-apoptotic forms of programmed cell death, i.e., autophagy, necroptosis, ferroptosis. Thus, pioneering studies aimed at inducing or manipulating autophagy, necroptosis, ferroptosis are under investigation for the final goal to eliminate apoptosis-resistant cancer cells, which still survive following several rounds of chemotherapy. These approaches might represent an alternative way to overcome tumor apoptotic resistance and chemo-resistance of cancer cells, and might provide additional promising therapeutic anticancer strategies. 

### 6.2. Autophagy

Autophagy is a catabolic process, highly conserved in all eukaryotes, through which cells remove and recycle into the double-membrane vesicles “autophagosomes” intracellular components, such as unused proteins, damaged organelles, and intracellular pathogens, that are degraded by lysosomal proteases [217]. The molecular process underlying autophagy, summarized in Figure 5, is complex, and is finely regulated by production of ATP and by the mTOR and AMP kinase pathways, which are linked to energy metabolism [218].

Autophagy mainly functions for the regulation of the turnover of cellular organelles, but it can also contribute to cell death. A close relationship between apoptosis and autophagy has been described, where autophagy controls apoptosis and vice versa. Induction of apoptosis or initiation of autophagy is determined by Bcl-2 proteins that control the balance of pro and anti-apoptotic signals and the first steps of autophagy. Autophagy is also recognized as a cellular and mammalian survival mechanism activated during stress, for example microbial infections or nutrient deprivation. Indeed, cellular stresses, including amino acid deprivation and ER stress, or signals that inhibit mTOR pathway can induce the transcriptional activation of autophagic genes or post-translational modifications, promoting autophagy [219]. Therefore, dysregulation of autophagy leads to development of different diseases, including cancer.

The role of autophagy in cancers was the focus of intense research and is still under investigation, due to the observed multifaceted and sometimes apparently opposite nature of autophagy in cancer. Indeed, researchers described a tumor suppressive role of autophagy at early stages of the tumor, while a tumor-promoting role was observed at later stages, rendering it still debated whether autophagy can be considered a valid or limiting therapeutic target. Autophagy was first described as a tumor-suppressive agent. Evidence reported that mutations in oncogenes or tumor suppressor genes, by inducing their activation or inhibition, respectively, led to suppression of autophagy, and in turn to inhibition of tumor growth. Moreover, partial or complete inhibition of autophagy through deletion of the autophagic genes Beclin1, Atg5, and Atg7, promoted development of various neoplasia [220,221,222]. However, a great deal of data supported a pro-tumor role for autophagy. Autophagy was associated with cancer cell migration, invasiveness, metastasis formation, and resistance to chemo- and radiotherapy. For this reason, the autophagic process can be considered a marker for tumor diagnosis as it is involved in tumor growth and metastatic progression. Interestingly, extracellular hostile conditions, such as hypoxia, low nutrients and growth factors, ROS, and lactate, typically present in the tumor microenvironment, were able to induce activation of autophagy, promoting cancer cell survival and tumor growth [223,224,225,226]. Thanks to the prominent role of autophagy in the regulation of mitochondria turnover, and to the primary role of mitochondria in energy production, studies are emerging to evaluate autophagy contribution to the tumor metabolism. These studies shed a light for a better understanding of the altered metabolic phenotype of tumors and increase the knowledge on how the TME sustains growing tumors. Findings correlated the metabolic reprogramming of tumor cells with autophagy activation. In fact, knock down of respiratory chain and complex III or inhibition of mitochondrial OXPHOS, by inducing mitochondrial dysfunctions, either slowed down the production of ATP, which led to accumulation of AMP and activation of AMPK, or enhanced *ATG* genes transcription, in turn promoting autophagy and tumor cell survival [227,228]. Moreover, a role for autophagy in sustaining glycolysis and amino acid metabolism was reported in breast and pancreatic cancers, hematological malignancies, and Ras-driven tumors [219,229,230,231,232]. In all these tumors, autophagy was fundamental for a correct functioning of mitochondria. Indeed, inhibition of autophagy impaired OXPHOS and enhanced mitochondrial dysfunctions. 

Autophagy also participates in the modulation of the metabolic crosstalk between tumor and non-tumor cells, to sustain tumor growth [233,234]. For instance, it was observed that in oxidative cancer cells autophagy is induced by the oxidation of lactate, leading to cancer cell survival and tumor growth [225]. Interestingly, a recent work highlighted that autophagy had a role not only in the intrinsic metabolism of tumor cells, promoting their growth, but also in that of non-tumor cells in a non-cell autonomous manner, in order to further sustain and promote the growth and progression of tumor cells. In more details, in pancreatic cancer, the metabolic support to cell growth was reported to be strongly depending on autophagy induced in non-tumor pancreatic stellate cells [235]. The authors observed that in conditions of hypoxic and nutrient poor supply, when pancreatic cancers were co-cultured with pancreatic non-tumor stellate cells, they induced somehow upregulation of autophagy in their non-tumor counterpart, leading to the secretion of the non-essential amino acid alanine, in support and sustain of their mitochondrial metabolism [235,236]. Consequently, adjacent pancreatic tumor cells up-took more alanine that fueled TCA cycle and was used for lipid neo-synthesis [235]. In the same vein, inhibition of autophagy in pancreatic non-tumor stellate cells restrains expression of the pro-migratory cytokine IL-6 and of the extracellular matrix proteins in pancreatic cancer, limiting tumor migration and invasion [237]. Taking into account these considerations, approaches aimed at blocking autophagy in pancreatic stellate cells could represent promising alternative treatments in pancreatic cancer.

Oxidative stress and ROS production in cancers can also activate autophagy through different pathways, including inhibition of mTORC1 or NFkB [238,239]. Activation of autophagy by ROS was associated with tumor progression versus malignancy. In this case, autophagy played a role during extracellular matrix detachment in human breast ductal carcinoma [240], and in chemo- and radio-resistance in bladder cancer cells and in lung adenocarcinoma, respectively [241,242].

In accordance with this evidence, different autophagy inhibitors have been proposed as potential anticancer agents, which showed different degrees of efficacy. One of the first used in clinical trials was hydroxychloroquine (HCQ) and its derivatives and alternatives, which target lysosome function [243,244,245]. Additional anti-autophagy molecules evaluated were the inhibitors of the class III phosphatylinositol 3-kinase (Vps34) [246], inhibitors of the autophagy initiating kinases ULK1 [247], and inhibitors of the cysteine ATG4B [248,249]. These inhibitors were also tested in combination with classical anti-tumor drugs. They showed good efficacy in inhibiting autophagy, and in blocking tumor growth in vitro, and in sensitizing to anti-tumor therapy [247,249,250,251,252]. However, they are still under exploitation to test their selectivity and toxicity.

Autophagy can also become lethal, or facilitate other forms of cell death, including apoptosis. Autophagic cell death is characterized by the presence of autophagosomes, by a massive vacuolation of the cytoplasm, and is governed by the canonical autophagic pathway [173]. In the case of resistance to apoptosis, frequent in many cancers, targeting autophagic cell death can therefore represent an alternative therapeutic strategy [253].

The induction of lethal autophagy, indicated by some authors as abortive [254], has been described for some natural and synthetic compounds, and for some drugs in different types of cancers [254,255,256,257]. For example, salinomycin has been reported to cause autophagic death in glioblastoma cells through a ROS-dependent mechanism [254]. The development of autophagic death activators therefore represents a great potential for anti-neoplastic chemotherapy advances.

### 6.3. Necroptosis

Necroptosis is a new form of programmed caspase-independent cell death, with features very similar to necrosis [258,259]. Necroptosis is triggered through a series of stimuli, including the ligands of the death receptors, toll-like receptors (TLRs), and interferon (IFN) receptors, and is induced by inhibition of caspase-8 and expression of receptor-interacting protein (RIP) kinase, in particular RIPK3 [260]. It can be inhibited either with Necrostatin-1 (Nec-1), by blocking the activity of the RIPK1 receptor, or with GSK872 which inhibits RIPK3 [261,262].

The molecular mechanisms leading to necroptosis execution are complex and can be summarized as above. Following the binding between the death ligand and tumor necrosis factor receptor 1 (TNFR1), complex I is formed by recruitment of the TNFR-associated death domain (TRADD), RIPK1, TRAF2. Within this complex, RIPK1 is polyubiquinated and activates in turn the NFkB pathway, and cell survival prevails. If RIPK1 is deubiquinated, the signal switches towards the activation of cell death pathways, and in particular versus the activation of apoptosis or necroptosis. Switch between these two cell death modes strongly depends on activation of caspase-8 within the complex II. Indeed, active caspase-8 inactivates RIPK1 and RIPK3, in favor of apoptosis execution. Pharmaceutical or genetic inhibition of caspase-8 instead supports RIPK1 interaction with RIPK3 and their subsequent activation within the necrosome complex, directing cell death to necroptosis. Activation of RIPK3 leads to a series of cascade signaling events culminating with the extracellular release of cellular content, and subsequent activation of inflammatory response [260]. For more details, see Figure 5.

Recent findings reported that necroptosis could have both a pro-tumoral and a tumor suppressive role during the onset of cancer development, strongly depending on the type of cancer, stage, and grade [261,262,263]. These roles of necroptosis are not the focus of this review and will not be discussed herein. Like the other forms of cell death, necroptosis was investigated in cancer therapy as well, in order to develop alternative approaches to fight against apoptotic resistance. 

As for apoptosis, autophagy, and ferroptosis, an interrelationship between necroptosis and the metabolism has been observed in cancer cells. Little is still known about the molecular mechanism at the basis of necroptosis modulation by tumor metabolism. Findings reported that RIPK3 could activate enzymes in metabolic pathways, in particular glycogen phosphorylase (PYGL) [264], and pyruvate dehydrogenase (PDH) [264]. This leads to an increase in aerobic respiration. In addition, the activity of glutammolysis ammonia ligase and glutamate dehydrogenase-1 are influenced by RIPK3 [264,265]. 

RIPK1/3 immunoprecipitation and ^32^P kinase experiments performed under hypoxic and glucose free conditions in human colorectal cancers showed enhanced activation of RIKP1/3, and production of mitochondrial superoxide, strongly favoring necroptosis [266]. Glucose addition to hypoxic cells inhibited necroptosis by inducing increase in GLUT-1 and GLUT-4 expression, intracellular ATP, pyruvate, and lactate levels [267]. By contrast, treatment aimed at inhibiting GAPDH or mitochondrial pyruvate carriers confers resistance to the necrotic pathway [266]. Interestingly, recent findings reported that tumor protection from necrotic cell death under glucose starvation conditions involves the p53-regulated long noncoding RNA, TRINGS. Indeed, TRINGS control necroptosis by inhibition of the STRAP–GSK3β–NF-κB pathway [267].

Interestingly, hyperglycemia and an enhanced glucose uptake by cancer cells were shown to regulate the interplay between apoptosis and necroptosis, in favor of necroptosis activation. It has been reported that in presence of high concentration of glucose, human primary T cells and monocytes shift versus RIP1-mediated necroptosis, instead of undergo extrinsic apoptosis [268].

The action of drugs against the necroptotic pathway could be a new approach for cancer therapy, bypassing the apoptotic process [262]. Specifically, chemotherapeutic agents, including members of the tumor necrosis factor receptor (TNFR) superfamily, pattern recognition receptors (PRRs), T cell receptors (TCRs), or natural compounds can be used to kill cancer cells by inducing or manipulating this caspase-independent mechanism [269].

### 6.4. Ferroptosis

Ferroptosis is a novel form of iron-dependent, non-apoptotic regulated cell death, distinct from the other forms of programmed cell death mentioned above. It is characterized by reduced mitochondrial size and higher mitochondrial membrane density, disruption of outer mitochondrial membrane and a substantial reduction in mitochondrial cristae, leading to the oxidative membrane damage [270,271,272]. It is induced when the metabolic balance between intracellular levels of ROS and the antioxidant activity of glutathione-dependent peroxidase 4 (GPX4) switches versus an exacerbate ROS and phospholipid hydroperoxides production and accumulation [273]. 

Different stimuli can activate the ferroptotic pathway. Indeed, cystine depletion, or inhibition of the amino acid antiporter system xc− by Erastin, or inactivation of GPX4 by RSL3, as well as the activation of the iron transporters transferrin and lactotransferrin. Additionally, metabolic stresses, such as nutrient depletion, especially cystine, hypoxia, and radiation can induce ferroptosis. Iron, together with glutamate and glutamine, as well as the mevalonate play an essential role in the modulation of this type of cell death. Induction of cytosolic ferritin degradation through autophagy, the so-called “ferritinophagy” process, also promotes ferroptosis [274]. On the contrary, inhibition of the ferroptotic pathway occurs following treatment with ferrostatin-1, the iron chelator deferoxamine (DFO), or the antioxidant N-acetyl-cysteine (NAC) [275].

Accumulating experimental evidence correlated ferroptosis with some metabolic pathway. Indeed, mitochondrial respiration, lipid and amino acid metabolism, all inducing overproduction of ROS, can promote ferroptosis. Moreover, an excessive autophagic induction, which favors an enhanced protein degradation, provides iron and lipid accumulation which triggers lipid peroxidation, and in turn ferroptosis [276,277,278,279]. Iron by itself promotes ferroptosis through the lipid peroxide-generating Fenton Reaction [280]. 

Usually, non-tumor cells can die through ferroptosis. Dysregulation of this process was associated with various pathological conditions, including cancers [277]. In this regard, literature data demonstrated a tumor suppressor efficacy for ferroptosis [281]. Ferroptosis occurs in cancer cells by activation of the RAS signaling pathway, and through inactivation of both the system xc− and GPX4, which in turn leads to accumulation of ROS and iron, thus culminating in lipid peroxidation and cell death.

Several proteins and genes are involved in ferroptosis induction and execution. Two key players in its modulation in cancer cells are the tumor suppressor p53 and iron. Regarding p53, experimental studies showed that it can promote ferroptosis by inhibiting the transcription of the SLC7A11 gene, leading to impairment of both cystine uptake and GPX4 activity [282]. Studies in carcinoma cells reported that the decrease in GPX4 activity was accompanied by an increase in the production of lipid ROS [281]. Moreover, repression of SLC7A11 induced by p53 promoted ferroptosis through activation of lipoxygenases activity [283]. P53 was also able to induce ferroptosis by inhibiting the transcription of squalene and ubiquinone, two known anti-ferroptotic agents, leading to inhibition of the mevalonate pathway [284].

Iron plays a peculiar role in several biological processes, due to its presence in the prosthetic group of enzymes and proteins, such as heme, and to its participation in enzymatic reactions (heme synthesis). Part of the iron internalized by the cells is stored within ferritin, whereas the cytoplasmic remaining free iron catalyzes free radical formation via Fenton Reaction. This implies an iron-mediated enhanced ROS production, leading to oxidative damage, dangerous for the cells. Evidence in the literature shows that iron represents an essential element and a modulator of ferroptosis. Indeed, an iron overload by increasing the intracellular iron pool strongly favors the ferroptotic death pathway [285]. On the contrary, silencing or repression of the iron specific receptor (TFR1) significantly inhibits ferroptosis [217,282]. Moreover, treatments aimed at inhibiting iron levels through the use of the iron chelators, DFO, or the lipophilic antioxidants vitamin E and ferrostatin-1 blocked or hindered the ferroptotic process [286]. Therefore, all these studies suggested that imbalances between intracellular free and redox iron in favor of iron overload, and induced by changes in ferritin expression levels, are peculiar for activation of ferroptosis [287].

Considering the regulatory role of mitochondria in OXPHOS and iron metabolism, these organelles have also been studied regarding ferroptosis. Experimental data demonstrated their role in ferroptosis induction, especially in ferroptosis promoted by cystine deprivation, and related the mitochondrial and iron metabolism to this type of cell death [287]. First, mitochondrial morphological changes characterize this form of cell death. Secondly, evidence associated mitochondrial metabolism to increased cellular ROS production, as well as mitochondrial damages occurring during ferroptosis and enhanced mitochondrial ROS levels [282,287]. Moreover, several enzymes of the TCA cycle and of the ETC are involved in ferroptosis [277,287]. Lastly, several proteins able to regulate the iron transport across the mitochondrial membrane (i.e., mitoferrin 1 (Mfrn1) and mitoferrin 2 (Mfrn2)), the voltage-dependent anion channels (VDACs), or the mitochondrial ferritin that regulate mitochondrial iron accumulation were linked to ferroptosis [282,288,289]. All of this evidence conveys to the hypothesis that ferroptosis modulation could represent a potential novel approach in anti-cancer therapy, especially for those cancers resistant to chemo- and/or radiotherapy [281]. Several molecules with a known pro-ferroptotic ability are under exploitation for their potential use in therapy, and still need further studies for their optimization and validation for clinical application. Among them, RSL3, Erastin, Sorafenib, Sulfasalazine, and known chemotherapeutic drugs, i.e., Cisplatin [290], Altretanine [291], and Artesunate [292] showed efficacy in promoting ferroptosis. In hematological malignancies and in gastric cancers, ferroptosis induction by erastin seems to involve the c-JUN N-terminal kinase (JNK), p38-MAPK, and the RAS-RAF-MEK pathways [293]. In addition, studies reported that cisplatin could induce ferroptosis through depletion of intracellular GSH [290]. Interestingly, the cytotoxic effect of cisplatin was enhanced both in vitro and in vivo in combination treatments with RSL3 or erastin [294]. Based on this, studies in cancers of different histotypes suggested that co-treatment with cisplatin and erastin or RSL3 worked synergistically in stimulating ferroptosis, consequently enhancing cisplatin cytotoxicity [290,294,295,296]. A similar trend was observed for paclitaxel, where the combination of paclitaxel with RSL3 by promoting ferroptosis through suppression of SLC7A11, demonstrated more efficacy in reversing chemoresistance than each treatment alone [297]. Lastly, in vitro and in vivo studies found that treatments combining erastin with radiotherapy might represent an additional strategy to increase sensitivity of cancer cells to radiation as well [297,298,299].

## 7. Conclusions

The view of cancer as a metabolic disease has gradually been replaced over the years by the view as a genetic disease and this has strongly influenced therapeutic approaches. Recently, also thanks to the growing importance of metabolomics and systems biology, there has been a renewed interest in the metabolism, not just energy, of cancer cells.

The remarkable metabolic adaptability of cancer cells can be understood as the ability to maintain a balance between energy production, biosynthesis of biological macromolecules, and conservation of the redox balance in non-permissive microenvironmental conditions, first of all the hypoxic one, for the survival of non-cancerous cells. Recently, substantial progress has been made in understanding the molecular events that regulate the metabolic switching of cancer cells [82]. The large amount of information accumulated so far clearly indicates the fundamental role played by HIF in this metabolic adaptation, which represents the winning strategy of the tumor cell.

The process of metabolic adaptation could therefore constitute the Achilles heel of the cancer cell, as long as it is dissected and understood from a biochemical and molecular point of view.

The development of highly specific and selective metabolic inhibitors, together with the deepening of knowledge of the mechanisms underlying the metabolic plasticity of cancer cells, could lead to a profound change in the therapeutic approach against cancer in the coming years. Provided, however, that we have clinical data describing the metabolic profiles of human tumors to determine which metabolic alterations are prevalent in specific tumor types.

Clarifying the mechanisms through which the metabolic plasticity of cancer cells is achieved may therefore represent a further arrow in the arch of clinicians to design personalized therapeutic pathways based on the peculiar metabolic characteristics of the individual tumor with a view of precision medicine.

## Figures and Tables

**Figure 1 biomedicines-09-01942-f001:**
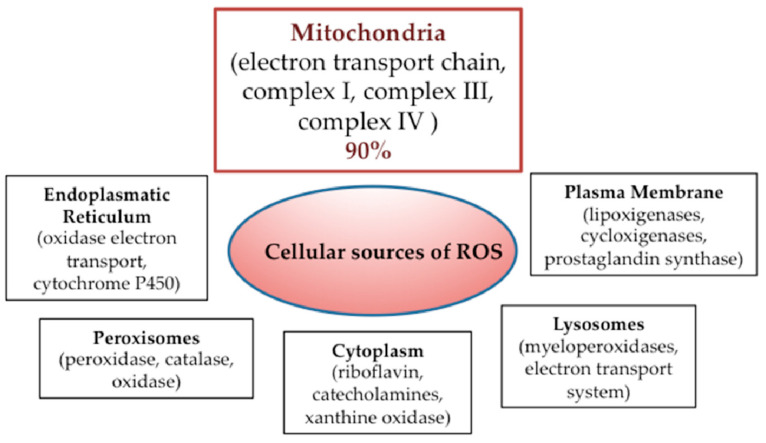
The main intracellular sources of ROS.

**Figure 2 biomedicines-09-01942-f002:**
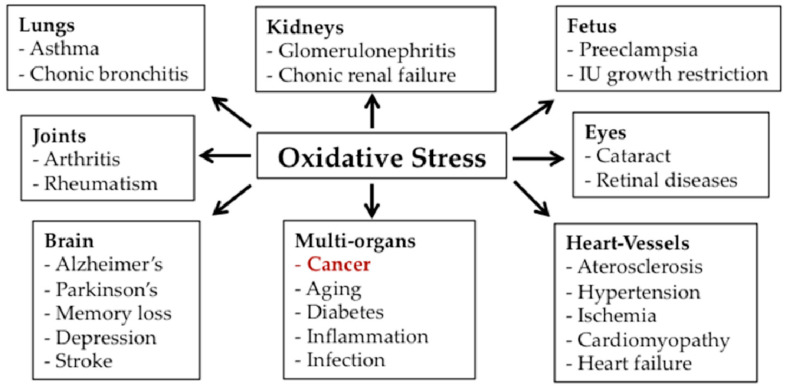
Oxidative stress represents a hallmark of several human diseases.

**Figure 3 biomedicines-09-01942-f003:**
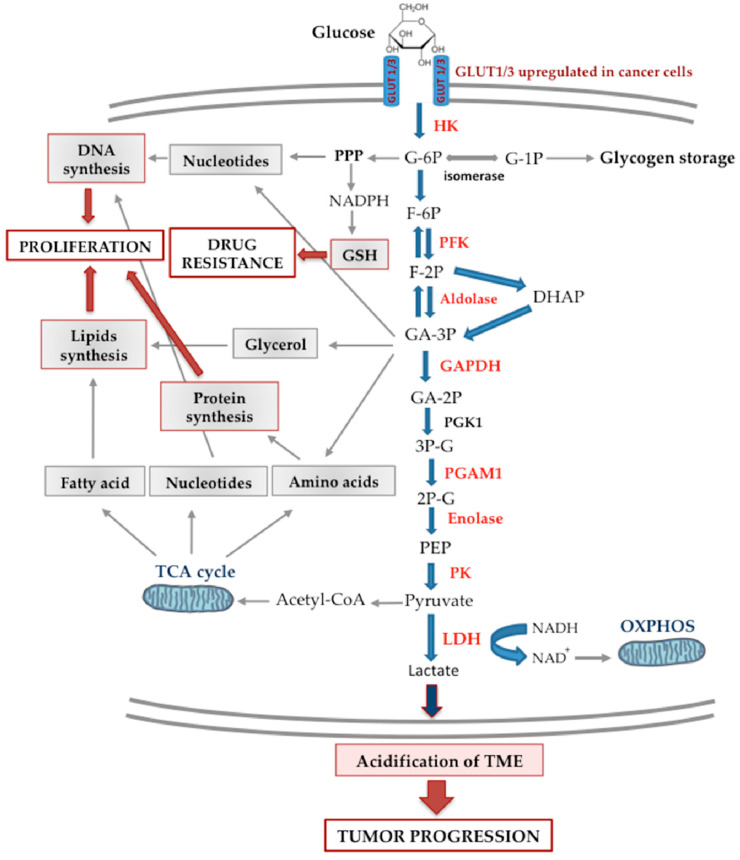
Glucose metabolism in cancer cells. The cell membrane glucose transporters (GLUT 1/3) mediate the glucose uptake, which is degraded to pyruvate by glycolysis. Under anaerobic and aerobic glycolysis (Warburg effect), pyruvate is converted to lactate, thereby regenerating NAD+ to supply the glycolytic processes. Pyruvate, into mitochondria, then undergoes oxidative phosphorylation by the tricarboxylic acid cycle (TCA), leading to the formation of ATP. Glycolytic enzymes are upregulated in many cancer types (marked in red), leading to enhanced generation of glycolytic intermediate, which functions as precursors for numerous metabolic pathways necessary for the biosynthesis of cellular components (grey boxes). Legend: G-6P, glucose-6-phosphate; F-6P, fructose-6-phosphate; DHAP, dihydroxyacetone phosphate; GA-3P, glyceraldehyde-3-phosphate; G-2P, 1,3-biphosphoglycerate; 2P-G, 2-biphosphate-glycerate; PEP, phosphoenolpyruvate; ATP, adenosine triphosphate; NAD+, nicotinamide adenine dinucleotide (oxidized form); NADH, nicotinamide adenine dinucleotide (reduced form); TCA, tricarboxylic acid; LDH, lactate dehydrogenase; PFK, phosphofructokinase; HK2, Hexokinase2; GDPH, glyceraldehyde-3-phosphate-dehydrogenase; PKM2, pyruvate kinase M2; PGAM1, phosphoglycerate mutase 1.

**Figure 4 biomedicines-09-01942-f004:**
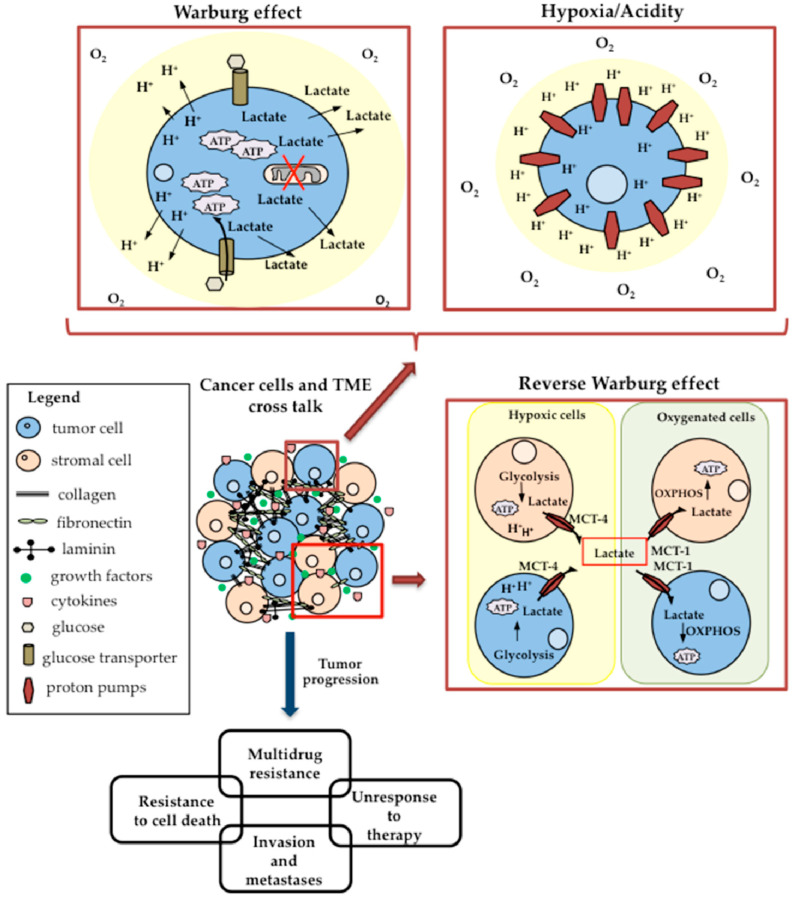
Cross talk between tumor cells and tumor microenvironment (TME). Normal cells, extracellular matrix proteins, growth factors, cytokines, metabolites, etc., characterize the TME. The interplay between tumor cells and their surrounding complex microenvironment can determine metabolic adaptation of cancer cells and tumor progression. Typical metabolic features of cancer cells are represented in boxes: Warburg effect, hypoxia, and acidity; reverse Warburg effect. Adapted from [82]. Legend: TME, tumor microenvironment; MCT-1, monocarboxylate transporter-1, MCT-4, monocarboxylate transporter-4; ATP, adenosine triphosphate; OXPHOS, oxidative phosphorylation.

**Figure 5 biomedicines-09-01942-f005:**
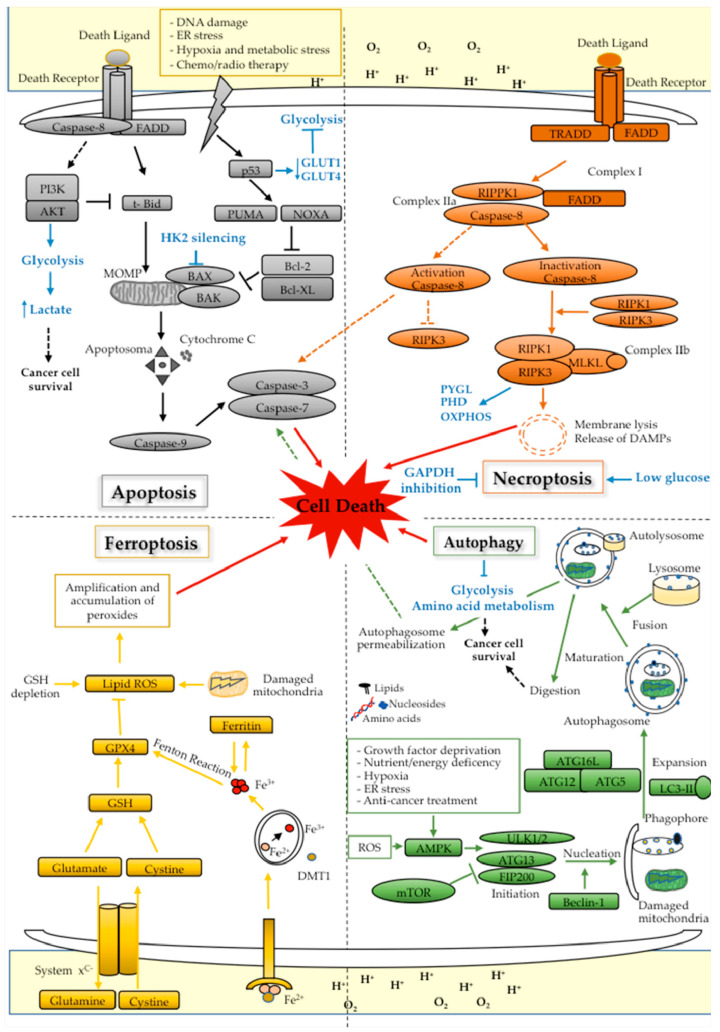
Biochemical pathways involved in different types of cell death. The scheme displays different pathways of cell death. A schematic representation shows the main molecules involved in the following pathways: Apoptosis (grey), Necroptosis (orange), Ferroptosis (yellow), and Autophagy (green). The text presented in blue indicates key molecules of some metabolic pathways that influence the processes of cell death or vice versa. Legend: ER, endoplasmic reticulum; FADD, FAS Associated protein with Death Domain; BCL-XL, B-cell lymphoma-extra large; BAX, BCL2 Associated X; BAD, BCL2 Associated Agonist Of Cell Death; t-Bid, truncated BH3 Interacting Domain Death Agonist; PI3K, Phosphoinositide 3-kinase; AKT, serine/threonine kinase; MLKL, Mixed lineage kinase domain-like protein; DAMPs, Damage-associated molecular patterns; DMT1, Divalent metal transporter-1; AMPK, AMP-activated protein kinase; mTOR, mammalian target of rapamycin; FLIP, Cellular FLICE-like inhibitory protein; LC3, Microtubule-associated protein 1A/1B-light chain 3; ROS, reactive oxygen species; OXPHOS, oxidative phosphorylation; GSH, reduced glutathione; Bcl-2, B-cell lymphoma 2; GLUT, glucose transporters; HK2, hexokinase 2; GAPDH, glyceraldehyde-3-phosphate-dehydrogenase; MOMP, outer mitochondrial membrane; PUMA, p53 upregulated modulator of apoptosis; ULK1/2, autophagy initiating kinases 1/2; RIP, receptor-interacting protein; TRADD, TNFR-associated death domain; PYGL, glycogen phosphorylase, GPX4, glutathione-dependent peroxidase 4.

## Data Availability

Not applicable.

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
