# Peer review of "Targeting the Interplay between Cancer Metabolic Reprogramming and Cell Death Pathways as a Viable Therapeutic Path"

_biomedicines, 2021, doi:10.3390/biomedicines9121942_

Round 1

Reviewer 1 Report

In this review, Iessi et al., discuss the metabolic reprogramming that occur in cancer, and their therapeutic perspectives. They introduce cancer as a metabolic disease fueled by oxidative phosphorylation and glycolysis. They then review the role of the tumor micro-environment, as well as the metabolic regulation of apoptosis, necroptosis and ferroptosis, three modes of cell death.

I found that the review was relatively complete and well-written. I can see how it will help students and researchers new to the field. I am thus supporting its publication.

I noticed a few typos and items that need clarification:

- Line 59: complexes I and II (respectively)
- The figure legend of Fig. 1 and Fig. 2 has been swapped
- Line 65: “Recent literature data highlighted as mitochondrial metabolism” doesn’t sound right to me.
- Fig. 1: Cytoplasm (not Cytiplasm)
- Line 155: Electron in singular?
- Line 122: I would define PPP as an alternative to glycolysis (parallel maybe)
- Line 145: Reference is missing
- Line 152-174: There are multiple enzymes in glycolysis and they should be mentioned e.g. not only HK2 is an hexokinase, HK1 as well, depending on the tissue.
- Line 163: DHAP should be mentioned
- Line 168: how about PKM M1? There is some confusion between M1 and M2 isoforms of PKM that needs to be clarified. PKM2 is not a gene name.
- Fig. 3: add DHAP
- Line 218: the exact number of ATP made by OXPHOS is still debated in the field. Perhaps mention that it can be 31, or 36, or 38.
- Fig. 4: doesn’t look good when printed. I notice this is a common problem at MDPI, at least for the reviewer. Make sure the final version will be of higher quality.
- Fig. 5: I think it confusing (wrong) to see a positive arrow between anti-apoptotic proteins (BLC2, BCL-XL) and BAX/BAK. Similarly, tBid actives BAX and BAK, not directly MOMP.
- Fig. 5: the system xc- imports cystine, not glutammine (with only one m by the way).
- In general there is a confusion between cystine (dimer of cysteine) and cysteine in the whole chapter on ferroptosis. SLC7A11 imports cystine (not cysteine).
- Lines 698-709: this whole paragraph lacks references.

Author Response

Reviewer 1

In this review, Iessi et al., discuss the metabolic reprogramming that occurs in cancer, and their therapeutic perspectives. They introduce cancer as a metabolic disease fueled by oxidative phosphorylation and glycolysis. They then review the role of the tumor micro-environment, as well as the metabolic regulation of apoptosis, necroptosis and ferroptosis, three modes of cell death.

I found that the review was relatively complete and well-written. I can see how it will help students and researchers new to the field. I am thus supporting its publication.

I noticed a few typos and items that need clarification:

Authors

We thank this reviewer for the positive comments. We also welcome her/his suggestions.

Reviewer

Line 59: complexes I and II (respectively)

Authors

We have completely rewritten the paragraph (highlighted in yellow) and the inaccuracy has been removed.

Reviewer

The figure legend of Fig. 1 and Fig. 2 has been swapped.

Authors

We apologize for this inaccuracy. We modified the figure legends.

Reviewer

Line 65: “Recent literature data highlighted as mitochondrial metabolism” doesn’t sound right to me.

Authors

We thank the referee for her/his suggestion. In the revised version the sentence has been modified.

Reviewer

Fig. 1: Cytoplasm (not Cytiplasm)

Authors

We apologize for this typing error, which in the revised version of the figure has been corrected.

Reviewer

Line 155: Electron in singular?

Authors

We apologize for this typing error, which has been removed in the revised version of the paper.

Reviewer

Line 122: I would define PPP as an alternative to glycolysis (parallel maybe)

Authors

We thank the referee for her/his suggestion.

Reviewer

Line 145: Reference is missing

Authors

In the revised version of the paper a proper reference has been added.

Reviewer

Line 152-174: There are multiple enzymes in glycolysis and they should be mentioned e.g. not only HK2 is an hexokinase, HK1 as well, depending on the tissue.

Authors

We thank the reviewer for her/his helpful suggestion, which we took up by editing the paragraph also mentioning the other enzymes.

Reviewer

Line 163: DHAP should be mentioned

Authors

We thank the referee for his/her useful suggestion that has been accepted.

Reviewer

Line 168: how about PKM M1? There is some confusion between M1 and M2 isoforms of PKM that needs to be clarified. PKM2 is not a gene name.

Authors

We thank the reviewer for this suggestion. In the new version of the paper we have expanded the information related to the different types and forms of PK and their roles in tumor progression. The topic is so vast and interesting that it deserves to be treated in a dedicated paper.

Reviewer

Fig. 3: add DHAP

Authors

We appreciated this suggestion. The Figure 3 has been modified accordingly.

Reviewer

Line 218: the exact number of ATP made by OXPHOS is still debated in the field. Perhaps mention that it can be 31, or 36, or 38.

Authors

We thank the referee for his useful suggestions. This information has been included in the new version of the paper.

Reviewer

Fig. 4: doesn’t look good when printed. I notice this is a common problem at MDPI, at least for the reviewer. Make sure the final version will be of higher quality.

Authors

We thank the referee for her/his suggestion. In the revised version of the paper we have inserted the figures with the highest possible resolution.

Reviewer

- Fig. 5: I think it confusing (wrong) to see a positive arrow between anti-apoptotic proteins (BLC2, BCL-XL) and BAX/BAK. Similarly, tBid actives BAX and BAK, not directly MOMP.

- Fig. 5: the system xc- imports cystine, not glutammine (with only one m by the way).

Authors

We appreciated this suggestion. The Figure 5 has been modified accordingly.

Reviewer

- In general there is a confusion between cystine (dimer of cysteine) and cysteine in the whole chapter on ferroptosis. SLC7A11 imports cystine (not cysteine).

Authors

We apologize for this typing error, which in the revised version of the paper has been corrected.

Reviewer

- Lines 698-709: this whole paragraph lacks references.

Authors

We apologize for the error. In the revised papers the references has been added.

Reviewer 2 Report

This review article attempts to summarize the current state of knowledge based on studies that targeting the metabolic peculiarities of cancer could represent an innovative therapeutic strategy for the treatment of cancers. In this review the authors described the main metabolic differences between cancer and non-cancer cells and how these can affect the various cell death pathways, effectively determining the susceptibility of cancer cells to therapy-induced death. This is a well-written review about the metabolic reprogramming of cancer. Please see the comments below for more details:

1. In section 6 “Cell death and metabolism in cancer”, the authors should describe the molecular mechanisms about the relationship between the different types of regulated cell death and metabolism in cancers. However, there are a few molecular mechanisms about metabolism in Figure 5.

2. In all sections, the authors should add more information about targeting the metabolic peculiarities of cancer to apply a therapeutic strategy for the treatment of cancers.

3. All figures were with low pixel. The authors should change the high resolution of all figures.

Author Response

Reviewer 2

This review article attempts to summarize the current state of knowledge based on studies that targeting the metabolic peculiarities of cancer could represent an innovative therapeutic strategy for the treatment of cancers. In this review the authors described the main metabolic differences between cancer and non-cancer cells and how these can affect the various cell death pathways, effectively determining the susceptibility of cancer cells to therapy-induced death.

This is a well-written review about the metabolic reprogramming of cancer. Please see the comments below for more details:

Authors

We thank this reviewer for her/his positive comments. We also welcome your suggestions.

Reviewer

  1. In section 6 “Cell death and metabolism in cancer”, the authors should describe the molecular mechanisms about the relationship between the different types of regulated cell death and metabolism in cancers. However, there are a few molecular mechanisms about metabolism in Figure 5.

Authors

We have modified Figure 5 based on this suggestion.

Reviewer

  1. In all sections, the authors should add more information about targeting the metabolic peculiarities of cancer to apply a therapeutic strategy for the treatment of cancers.

Authors

Ok, as requested, further information on the therapeutic opportunities offered by the metabolic specificities of tumors has been inserted through the text.

Reviewer

  1. All figures were with low pixel. The authors should change the high resolution of all figures.

Authors

All the figures have been replaced with others of higher resolution.

Reviewer 3 Report

The review by Iessi et al. entitled metabolic reprogramming addresses very current issues whose interest is reinforced by the potential therapeutic consequences. Many reviews have been published recently on the subject which is very extensive. The difficulty is not to fall into the trap of wanting to do an extensive review but rather to focus on newer aspects of the cancer metabolism issue.

Strength : The most interesting point of this review is how tumor metabolism can influence different cell death pathways. It deserves much detail. The title of the review is too vague and should include the influence of metabolism on cell death.

Weaknesses : This review mixes too basic generalities (textbook level) about metabolism (for example OXPHOS (l.48-60) or glycolysis and Warburg effect) with more current and relevant points on the topic. This results in a text that is far too long and loses its attractiveness for the reader. In particular the chapters are too long and often not appropriate. For example, the chapter on mitochondria and cancer metabolism only talks about ROS which is far from being the only influence of mitochondria in tumorigenesis or resistance to treatments.

I am more concerned about a number of inaccuracies on various metabolic points:

  • For instance, l61-62 the explanation of the formation of ROS by the mitochondria is very vague and does not correspond to what we know about the mechanism and the reference 4 is not appropriate.
  • the authors must explain to me why they consider glycolysis as a mitochondrial function (l 133°)
  • the authors must imperatively review their definition of aerobic and anaerobic glycolysis and fermentation. It is not accurate and very confusing (l168-174)
  • L 247 The authors should clarify what they mean by oxidative phosphorylation of pyruvate, this term is not classical and seems to come from a confusion between mitochondrial oxphos and pyruvate oxidation by mitochondrial pyruvate dehydrogenase (without phosphorylation)

To conclude, this review does not become really stressful until chapter 6. This section should be rewritten with more detail and emphasis on the mechanistic and translational aspects of potential therapeutic possibilities.

Author Response

Reviewer 3

The review by Iessi et al. entitled metabolic reprogramming addresses very current issues whose interest is reinforced by the potential therapeutic consequences. Many reviews have been published recently on the subject, which is very extensive. The difficulty is not to fall into the trap of wanting to do an extensive review but rather to focus on newer aspects of the cancer metabolism issue.

Authors

We want to thank this reviewer for her/his comments and suggestions for improving our paper.

Reviewer

Strength: The most interesting point of this review is how tumor metabolism can influence different cell death pathways. It deserves much detail. The title of the review is too vague and should include the influence of metabolism on cell death.

Authors

We thank for these positive comments. We also welcome your suggestion to choose a title more appropriate to the content of our paper. The new title proposed is: “Targeting the interplay between cancer metabolic reprogramming and cell death pathways as a viable therapeutic path”

Reviewer

Weaknesses: This review mixes too basic generalities (textbook level) about metabolism (for example OXPHOS (l.48-60) or glycolysis and Warburg effect) with more current and relevant points on the topic. This results in a text that is far too long and loses its attractiveness for the reader. In particular the chapters are too long and often not appropriate. For example, the chapter on mitochondria and cancer metabolism only talks about ROS, which is far from being the only influence of mitochondria in tumorigenesis or resistance to treatments.

Authors

In accordance with what has been suggested, we have tried to summarize those paragraphs concerning the most well known topics and therefore considered too basic. We also added information on the impact of mitochondrial dynamics in the metabolic adaptation of cancer cells. We hope that in this way we have made it easier and more fluent to read this paper.

Reviewer

I am more concerned about a number of inaccuracies on various metabolic points:

For instance, l 61-62 the explanation of the formation of ROS by the mitochondria is very vague and does not correspond to what we know about the mechanism and the reference 4 is not appropriate.

Authors

We apologize for the inaccuracy. In the new version of the paper, the paragraph has been completely rewritten and the reference 4 replaced with others that are actually more appropriate.

Reviewer

The authors must explain to me why they consider glycolysis as a mitochondrial function (l 133°)

Authors

We are sorry. That was not what we meant. The sentence has been reworded to remove the error and misunderstanding.

Reviewer

The authors must imperatively review their definition of aerobic and anaerobic glycolysis and fermentation. It is not accurate and very confusing (l 168-174)

Authors

We apologize for the confusion and inaccuracy. The paragraph has been completely revised.

Reviewer

L 247 The authors should clarify what they mean by oxidative phosphorylation of pyruvate, this term is not classical and seems to come from a confusion between mitochondrial oxphos and pyruvate oxidation by mitochondrial pyruvate dehydrogenase (without phosphorylation).

Authors

Thank you for this suggestion. The sentence was actually unclear. In the new version of the paper we made it clear that we mean the oxidation of pyruvate by OXPHOS.

Reviewer

To conclude, this review does not become really stressful until chapter 6. This section should be rewritten with more detail and emphasis on the mechanistic and translational aspects of potential therapeutic possibilities.

Authors

In accordance with this request, we have expanded the informations on the mechanistic aspects and on the potential therapeutic implications.

Round 2

Reviewer 2 Report

This revised manuscript appears to have incorporated many of the suggestions of the previous reviewers and is improved. I have no further comment.

Author Response

Thanks to this reviewer for his encouraging comments 

Reviewer 3 Report

The authors have responded to most of my comments. However, there is still one point to be corrected. One cannot say : pyruvate oxidation by  OXPHOX. This is not correct : OXPHOS is related to the electron transport chain (OX) and ATP synthase (PHOS). I suggest to change to :mitochondrial oxidation of pyruvate.

Author Response

We thank the reviewer  for this important clarification. We have edited the text as suggested.